# Selective Oxidation of Clopidogrel by Peroxymonosulfate (PMS) and Sodium Halide (NaX) System: An NMR Study

**DOI:** 10.3390/molecules26195921

**Published:** 2021-09-29

**Authors:** Everaldo F. Krake, Wolfgang Baumann

**Affiliations:** Leibniz-Institut für Katalyse e.V., Albert-Einstein-Straße 29a, 18059 Rostock, Germany; everaldokiko@gmail.com

**Keywords:** clopidogrel, NMR study, oxone, peroxymonosulfate, sodium halide, thienopyridine

## Abstract

A selective transformation of clopidogrel hydrogen sulfate (CLP) by reactive halogen species (HOX) generated from peroxymonosulfate (PMS) and sodium halide (NaX) is described. Other sustainable oxidants as well as different solvents have also been investigated. As result of this study, for each sodium salt the reaction conditions were optimized, and four different degradation products were formed. Three products were halogenated at C-2 on the thiophene ring and have concomitant functional transformation, such as *N*-oxide in the piperidine group. A halogenated endo-iminium product was also observed. With this condition, a fast preparation of known endo-iminium clopidogrel impurity (new counterion) was reported as well. The progress of the reaction was monitored using nuclear magnetic resonance spectroscopy as an analytical tool and all the products were characterized by 1D-, 2D-NMR and HRMS.

## 1. Introduction

The direct insertion of halogens in (hetero)aromatic drugs, in a selective way, has been the object of much interest by the synthetic community [1]. The inclusion of a new C−X bond in these bioactive heterocyclic compounds can improve their physical and biological properties, increase potency, and be used as a handle in the further design and construction of pharmaceuticals [2]. Thiophene rings are five-membered heterocycles bearing sulfur atoms in their structure. Connected to a halogen, halothiophenes represent a class of bioactive molecules with extraordinary pharmacological properties [3,4], including the FDA-approved drugs Avatrombopag, Tioconazole, Lornoxicam, Rivaroxaban, and Brotizolam (Figure 1A).

In recent years, several methods of direct activation of halogens in organic compounds have been reported using safe halogen sources such as HX, NH_4_X and NaX (X = Cl, Br and I). To sustainably transform these halides into more reactive species, the use of oxidizing agents that conform to the principles of Green Chemistry is essential [1].

Like other green oxidants such as O_2_ and H_2_O_2_ [5], peroxymonosulfate (PMS, Oxone ^®^) has been widely used: (1) in the academia to develop new synthetic protocols [6]; (2) in pharmaceutical companies to promote oxidative stress testing of active pharmaceutical ingredients (API) to predict their degradation [7]; and (3) in hypersaline industrial wastewaters to remove organic contaminants [8,9,10]. This safe, sustainable, and inexpensive oxidant has shown extraordinary reactivity with alkaline metal halide salts [11,12,13,14,15,16,17,18,19,20,21,22,23,24,25,26,27] and with hydrogen halides (HCl, HBr and HI) [28,29,30].

In our previous work, we documented the reactive sequence of the oxidation–chlorination of Ticlopidine hydrochloride using PMS. We observed the formation of the reactive intermediate species (**DP-1**) containing a chlorine group at the C-2 carbon of the thiophene and the oxidation of cyclic amine to *N*-oxide in the piperidinic structure (Figure 1B) [31].

Clopidogrel hydrogen sulfate (**CLP**, Plavix ^TM^) is another thienopyridine drug that has powerful antiplatelet properties, and it plays an important role in the treatment of coronary, peripheral vascular and cerebrovascular diseases [32,33,34]. Several researchers have reported functional transformations of **CLP** using halogenated succinimide reagents (Figure 1C). Padi and coworkers have used *N*-bromosuccinimide (NBS) for the preparation of endo-iminium impurity **DP-2** on a large scale [35]. Jiao and coworkers have developed an efficient method using *N*-chlorosuccinimide (NCS) and dimethyl sulfoxide (DMSO) as a catalyst for the preparation of 2-Cl-clopidogrel **DP-3a** [36].

Continuing our efforts in the development and optimization of sustainable prediction methods for degradation of active pharmaceutical ingredients, we wanted to understand the oxidative halogenation reaction of heterocycles containing non-hydrohalic acids with safe halogen sources. Here, we show the oxidation of **CLP** using PMS/sodium halides (NaX, X = Cl, Br and I; Figure 1D). We observed the formation of four interesting classes of products that can predict the degradation of thienopyridines under high salinity media. To the best of our knowledge, detection, characterization, and selective preparation of the products in this oxidative method are first reported herein.

## 2. Results and Discussion

### 2.1. Impact of Halides on the Transformation of Clopidogrel

#### 2.1.1. Influence of Chloride

Our experimental work started with the optimization of conditions for the oxidative chlorination of **CLP** with sodium chloride (NaCl) as a chlorine source, and different oxidant agents (Table 1). We monitored the reaction by NMR and HPLC. In the initial experiments, the treatment of **CLP** with H_2_O_2_ or *tert*-butyl hydroperoxide (TBHP) in the presence of NaCl did not lead to the formation of products after 24 h at room temperature (Table 1, Entries 1 and 2). Using PMS in D_2_O, we observed the complete consumption of **CLP** and the formation of various degradation products after one minute. Inspired by our previous work [31], we decided to investigate the co-solvent effect for this reaction. We performed some experiments with dichloromethane-d_2_ (CD_2_Cl_2_), chloroform-d (CDCl_3_), dimethyl sulfoxide-d_6_ ((CD_3_)_2_SO), benzene-d_6_ (C_6_D_6_), and toluene-d_8_ (C_7_D_8_), but these deuterated co-solvents did not lead to the formation of any products. However, with acetone-d_6_, we observed the formation of an *N*-oxide product, **DP-****6** (48%), as well as the chlorinated products **DP-****3a** (4%) and **DP-****4a** (25%) after 5 h (Entry 4).

To investigate the selectivity between chlorinated products **DP-****3a** and **DP-****4a**, the same reaction was performed with other polar solvents (Entries 5–7): methanol-d_4_ (32%), THF-d_8_ (51%) and acetonitrile-d_3_ (51%). We observed increased formation of **DP-4a**, as well as small amounts of **DP-****3a**. With the optimized solvent in hand, we extended the method to optimize **DP-****3a** or **DP-****4a** using varying amounts of PMS and NaCl (Entries 8–13). We did not obtain **DP-****3a** with a conversion higher than 20%. Thus, our best conditions for **DP-****4a** formation used a mixture of D_2_O:CD_3_CN (2:1), PMS (2.0 equiv), and NaCl (2.0 equiv; Entry 8).

#### 2.1.2. Influence of Bromide

In parallel to the chlorination reaction, we submitted **CLP** to the oxidation process using NaBr as a bromine source (Table 2). We observed a quick conversion of the brominated products **DP-****3b** and **DP-****5b** with all solvents studied. Then, we extended our study varying the amount of PMS and NaBr. The best condition for the formation of **DP-****3b** was with D_2_O:CD_3_CN (2:1), PMS (0.5 equiv), and NaBr (1.0 equiv; Table 2, Entry 6). The combination of PMS (1.5 equiv) and NaBr (1.0 equiv; Table 2, Entry 8) provided **DP-****5b** in excellent yield.

#### 2.1.3. Influence of Iodide

When we used the iodide reagent NaI and other co-solvents (Entries 1–3), solubility problems were observed after the in situ oxidative conversion of the iodide to the reactive iodine species was generated. With NaCl, NaBr and NaI, acetonitrile-d_3_ was the most efficient deuterated co-solvent. Unlike with the halides described above, no substitution at the thiophene ring occurred (no formation of **DP-3c**, entries 4 and 5). Instead, we discovered a fast method (as compared to Padi’s) [35] to prepare **DP-2** in excellent yield by using the PMS/NaI system (Table 3, Entry 4).

### 2.2. Determination of Reaction Progress by NMR

#### 2.2.1. With NaCl

In Figure 2, a compilation of the ^1^H NMR spectra on the main functional transformations that occurred in the oxidative process, which resulted in the formation of the products in high yields, is shown. In addition, the data of the starting materials (as reference) and non-chlorinated intermediates **DP-****6** (mixture of diastereomers) are also included in this compilation of spectra, as described in Figure 2A,B [37]. The main observation among the NMR spectra of this work is the disappearance of the doublet (δ_H_ ~6.7 ppm, CD_3_CN) of proton H-3 of the thiophene ring and a conversion into a singlet in that same region which results from the insertion of a heteroatom at carbon-2 of this heterocycle (Figure 2C).

Figure 2C represents the progress of oxidative chlorination of **CLP** and NaCl after two hours (Table 1, Entry 12). In the first two hours of reaction, we observed the appearance of non-halogenated diastereomers of intermediate **DP-****6** with a new N^+^-OH bond, as well as the mono-chlorinated compound **DP-****3a**. As the reaction progressed towards the production of **DP-****4a**, the signals of **DP-****6** and **DP-****3a** decreased in the spectrum. This led us to conclude that the reaction pathway to obtain **DP-****4a** occurred via two simultaneous processes: oxidation-chlorination via **DP-****6** and chlorination-oxidation via **DP-****3a** (Figure 3). The final product, **DP-****4a,** is represented in Figure 4D, and bears a chlorine on the thiophene ring and an *N*-oxide in the piperidine structure. The mechanism of this reaction was shown in our previous work [31].

#### 2.2.2. With NaBr

Contrary to the PMS/NaCl process, the reaction with NaBr did not present *N*-oxide products, but a C-2 halogen/endo-iminium **DP-****5b** when PMS (1.0 equiv) was used (Figure 4B,D). To understand the development of this reaction, we started with 0.5 equivalent of PMS (Table 2, Entry 6; Figure 4A) and observed a mixture between **CLP** and mono-brominated product **DP-****3b**. In addition, we increased the amount of PMS to provide **DP-****5b** in a quantitative yield (Table 2, Entry 8; Figure 4D).

#### 2.2.3. With NaI

With NaI, we tried to obtain an iodinated product, **DP-3c,** under our experimental conditions. Instead, we observed the formation of endo-iminium **DP-****2** in high yield (Table 3, Entry 4; Figure 5B). Reducing the molar amounts of PMS, we detected a mixture between **CLP** and endo-iminium **DP-2** (Table 3, Entry 5, Figure 5A).

### 2.3. Characterization of the Products

Degradation products **DP-****2**, **DP-****3a**, **DP-****4a**, **DP-****3b** and **DP-****5b** were characterized directly from the reaction mixtures by HPLC-MS, HRMS, as well as by 1D- and 2D-NMR spectroscopy. Two-dimensional correlations ^1^H-^1^H COSY, ^1^H-^13^C HSQC and ^1^H-^13^C HMBC were used for the elucidation of the structure.

#### 2.3.1. Characterization of DP-2

Using NMR spectrometry (in CD_3_CN, see Table 4 and Appendix A), fifteen protons were detected, a value consistent with the molecular formula of **DP-****2**. In addition, the bidimensional ^1^H-^1^H COSY spectrum showed important correlations for structural elucidation of this molecule, including a cross peak between endo-iminium H-4 (δ_H_ 8.81, δ_C_ 162.40) and the singlet for H-10, which is part of a CH group (δ_H_ 6.39, δ_C_ 72.08). Shift data of the latter are similar to that observed for this position in the other molecules (Table 4). This feature proves that an endo-iminium compound, and not an exo-iminium one, is indeed formed. The edited HSQC exhibited the CH_2_ groups H-6 (δ_H_ 4.30–4.23 and 3.95–3.86, δ_C_ 49.94) and H-7 (δ_H_ 3.52–3.36, δ_C_ 24.04). With HRMS/ESI-TOF analysis, the molecular formula of **DP-2** was determined as C_16_H_15_ClNO_2_S^+^, with *m*/*z* calculated at 320.0517, and *m*/*z* observed at 320.0516 (−0.3 ppm error), indicating loss of one hydrogen atom in the molecule and the formation of the endo-iminium product. These analyses do not show a halogenation bond in the molecule and confirm the structure of this degradation product.

#### 2.3.2. Characterization of DP-3a

The molecular formula of product **DP-****3a** was established as C_16_H_11_O_2_NSCl_2_ by HRMS/ESI-TOF data in which *m*/*z* of 356.0279 was observed and was calculated for [M + H]^+^ 356.0288 (2.5 ppm error), indicating the insertion of a new chlorine atom into the molecule. In the NMR spectrum (in CD_3_OD, see Table 4 and Appendix A), the addition of the chlorine atom at carbon 2 resulted in an absence of the H-2 signal and the appearance of the singlet associated with H-3 (δ_H_ 6.50, δ_C_ 125.54) consistent with the molecular formula of **DP-****3a**. ^1^H-^13^C HSQC edited indicated the methylene groups: H-4 (δ_H_ 3.64 and 3.54, δ_C_ 51.07), H-6 (δ_H_ 2.98–2.84, δ_C_ 49.21) and H-7 (δ_H_ 2.76–2.70, δ_C_ 25.78). The homonuclear correlation spectroscopy (^1^H-^1^H COSY) showed a correlation with the protons H-3/H-4 and H-3/H-7. This analysis was important to explain the structure elucidation of **DP-****3a**. Also, ^1^H-^13^C HMBC correlations from proton signals for H-3, H-4, H-6 and H-10 were observed.

#### 2.3.3. Characterization of DP-4a

The oxidative chlorinated product **DP-****4a** was found to have the molecular formula C_16_H_15_O_3_NSCl_2_ by HRMS/ESI-TOF, in which *m*/*z* 372.0228 was observed and was calculated for [M + H]^+^ 372.0227 (−0.3 ppm error). This mass can also be seen in two retention times in the HPLC-MS chromatogram of Appendix A indicating the formation of two diastereomers with N^+^-OH bonds. In comparison to the data of ^1^H NMR with **CLP,** and based in our previous report about **DP-6** (Figure 2B,D in CD_3_CN; also see ref. [37]), it is possible to notice a significant change in the chemical shift in H-4 (δ_H_ 6.62 and 6.52 ppm; δ_C_ 62.50, 62.27 ppm) due to the asymmetric electric field (AMEF) generated in the molecule. This electrical field is the result of the polarization of the molecule through the dipole N^+^-OH generated in this oxidative process.

#### 2.3.4. Characterization of DP-3b

Similar to the results seen for the compound **DP-****3a**, the product **DP-****3b** showed the molecular formula C_16_H_15_O_2_NSClBr by HRMS/ESI-TOF, with *m*/*z* calculated 399.9774 and observed for [M + H]^+^ 399.9772 (−0.5 ppm error), specifying the insertion of a bromine atom into the molecule. In the NMR spectrum (in CD_3_OD, see Table 4 and Appendix A), the inclusion of a bromine atom at carbon 2 resulted in an absence of the H-2 signal and the appearance of a signal associated with H-3 (δ_H_ 6.70, δ_C_ 129.36), consistent with the molecular formula of **DP-****3b**. In addition, other changes in chemical shift in H-4 (δ_H_ 3.63 and 3.53, δ_C_ 51.10), H-6 (δ_H_ 2.95–2.80, δ_C_ 49.21) and H-7 (δ_H_ 2.77–2.74, δ_C_ 26.02) were noted. HMBC correlations from proton signals for H-3, H-4, H-6 and H-10 were observed.

#### 2.3.5. Characterization of DP-5b

Finally, in the NMR spectrum (in CD_3_OD, see Table 4 and Appendix A), fourteen protons were observed, a value consistent with the molecular formula of **DP-****5b**. In addition, HSQC showed the existence of an endo-iminium group in the piperidine moiety as a singlet (δ_H_ 9.05, δ_C_ 162.47), two methylene groups in H-6 (δ_H_ 4.44–4.37 and 4.00–3.91, δ_C_ 49.65), H-7 (δ_H_ 3.59–3.40, δ_C_ 24.38) and H-10 protons (δ_H_ 6.62, δ_C_ 72.72). HMBC correlations from proton signals for H-3, H-4, H-6 and H-10 were observed. Also, the molecular formula of **DP-****5b** was determined to be C_16_H_14_BrClNO_2_S^+^ by HRMS/ESI-TOF, with an observed *m*/*z* of 397.9622 and a calculated *m*/*z* of 397.9623 (0.3 ppm error) indicating the insertion of a bromine atom into the molecule, as well as the formation of the endo-iminium. This analysis corroborated with the data obtained from the NMR spectra.

## 3. Materials and Methods

### 3.1. Materials

Clopidogrel hydrogen sulphate (**CLP**) was kindly provided from RD&C Research, Development & Consulting GmbH, Vienna, Austria. Peroxymonosulfate (PMS, Oxone ^®^: 2KHSO_5_·KHSO_4_·K_2_SO_4_, MW = 614.74 g·mol^−1^) was purchased from TCI Deutschland. NaCl, NaBr and NaI were purchased from Sigma Aldrich, Inc. and used directly without further purification. All deuterated solvents were purchased from Deutero GmbH, Kastellaun, Germany.

### 3.2. Nuclear Magnetic Resonance Spectroscopy

All NMR spectra were recorded on a Bruker AVANCE III HD 400 MHz spectrometer at 297 K in D_2_O/CD_3_OD or D_2_O/CD_3_CN (2:1, *v*/*v*) solvents. 19 mg of **CLP** were dissolved in 0.6 mL of deuterated solvent mixture and used for ^1^H, ^13^C NMR, ^1^H-^1^H COSY, HSQC and HMBC analysis. Chemical shifts are reported in ppm (δ) and residual CD_3_OD (δ_H_ = 3.31 ppm, δ_C_ = 49.0 ppm) or CD_3_CN (δ_H_ = 1.94 ppm, δ_C_ = 1.32 ppm). Processing of the raw data was performed using Bruker TOPSPIN software.

#### 3.2.1. Recording of One-Dimensional NMR Spectra

The pulse conditions were as follows: ^1^H NMR, spectra (pulse sequence = zg30): number of data points (TD) = 43008, number of scans (NS) = 16, dummy scans (DS) = 2, spectra width (SWH) = 8012.820 Hz, acquisition time (AQ) = 2.6837 sec, spectrometer operating frequency (SFO1) = 400.13 MHz, π/2 pulse for ^1^H (P1) = 14.30 μs, relaxation delay (D1) = 1.27 s, line broadening (LB) = 0.10 Hz. ^13^C NMR spectra (pulse sequence = zgpg30): TD = 43702, NS = 256, DS = 2, SWH = 29411.766 Hz, AQ = 0.7429 sec, SFO1 = 100.626 MHz, LB = 1.00 Hz, D1 = 2.0 sec, P1 = 10.0 μs.

#### 3.2.2. Recording of Two-Dimensional NMR Spectra

^1^H/^1^H COSY (pulse sequence = cosygpppqf): TD = 2048 (F2), TD = 256 (F1) NS = 2, DS = 16, SFO1 = 400.132 MHz, D1 = 2.00 sec. ^1^H–^13^C HSQC (pulse sequence = hsqcedetgp): TD = 1024 (F2), TD = 256 (F1) NS = 2, DS = 16, SFO1 = 400.132 (F2) MHz, SFO1 = 100.622 (F1) MHz, D1 = 2.00 sec. ^1^H–^13^C HMBC (pulse sequence = hmbcgpndqf): The parameters were very similar to those used in the HSQC experiment.

### 3.3. Mass Spectrometry

The compounds were dissolved (about 0.05 mg·mL^−1^) in acetonitrile, using a solvent system of acetonitrile: formic acid, 0.1% in water [90:10, *v*/*v*] at a flow rate of 0.5 mL·min^−1^. The mass spectrum of the isolated products was acquired on a Xevo G2-XS Tof Mass Spectrometry instrument from Waters (Wilmslow, UK) in positive spray ionization (ES+) mode. The column used was an ACQUITY UPLC BEH C18 1.7 µm and with the following dimensions: 2.1 mm × 50 mm. The ES+ capillary was set at 3.0 kV, the source temperature at 120 °C and the desolvation temperature at 500 °C. Mass range was scanned between 50 and 750 amu.

### 3.4. General Description of the Experiment

To a glass vial with a solution of **CLP** (19 mg; 45.49 µmol; 1.0 equiv) in D_2_O/CD_3_CN (0.6 mL, 2:1 *v*/*v*) was added the corresponding halide salt (1.0 equiv) and PMS (2KHSO_5_·KHSO_4_·K_2_SO_4_, MW = 614.74 g·mol^−1^)—see Table 1, Table 2 and Table 3 for specific data. The solution was stirred before being transferred to an NMR tube and continuously monitored by ^1^H NMR at room temperature. After complete conversion, the reaction mixture was quenched with sodium thiosulfate and extracted with dichloromethane (3×). The organic layers were combined, dried over anhydrous MgSO_4_, filtered, and concentrated in vacuo. The desired halogenated products were then characterized without further purification. Characterization of the products by LC-MS is also possible directly from the reaction mixture.

## 4. Conclusions

In summary, we showed a selective transformation of clopidogrel hydrogen sulfate (**CLP**) by a PMS/halide system in aqueous acetonitrile media without employing a metal catalyst. With this method, we have prepared three major halogenated products using different halide salts. With this condition, a fast preparation of known endo-iminium clopidogrel impurity (**DP-2**, new counterion) was described as well. The new degradation products **DP-3a–b**, **DP-4a** and **DP-5b** were characterized using spectroscopic techniques (namely 1D-NMR, 2D-NMR and HRMS). We believe that this procedure is not only useful for generating clopidogrel derivatives but also very important to the study of drug degradation under hypersaline conditions. We are currently extending this study with other active pharmaceutical ingredients.

## Figures and Tables

**Figure 1 molecules-26-05921-f001:**
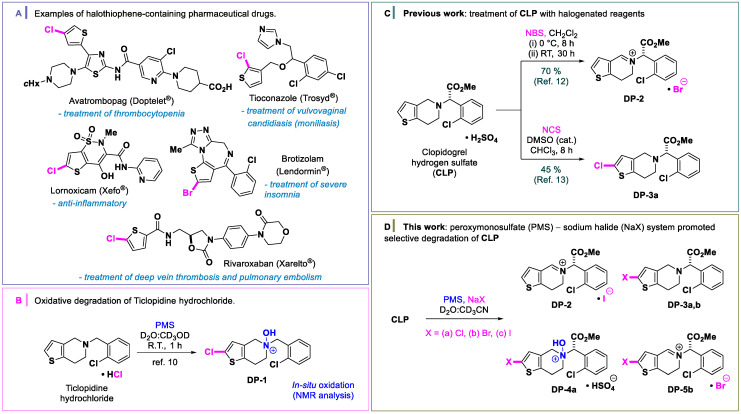
Halothiophene-containing pharmaceutical drugs and representative oxidation of **CLP**. (**A**) Examples of halothiophene-containing pharmaceutical drugs. (**B**) Our previous work: oxidative degradation of ticlopidine hydrochloride with PMS. (**C**) Previous work: treatment of **CLP** with halogenated succinimides. (**D**) This work: PMS-halide oxidation of **CLP**, anions may be halide or hydrogen sulfate.

**Figure 2 molecules-26-05921-f002:**
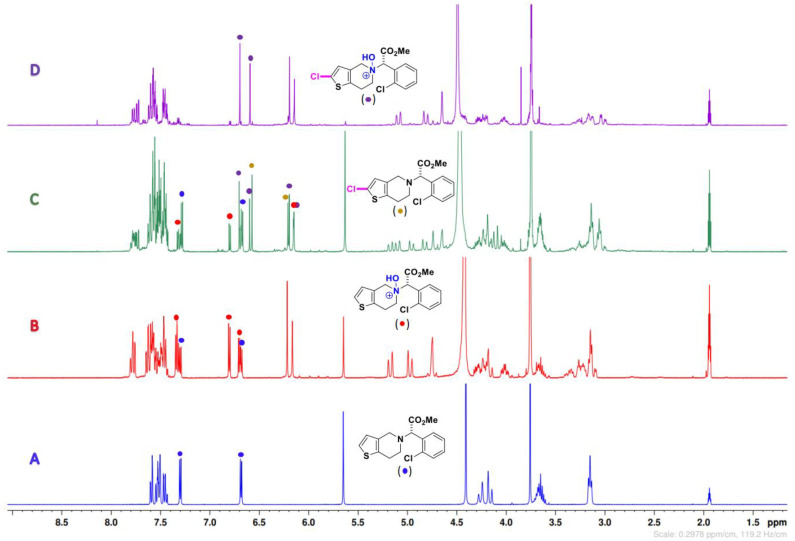
Compilation of ^1^H NMR spectra (400 MHz, CD_3_CN) of all compounds involved. Reaction conditions: (**A**) **CLP** (as ^1^H NMR reference); (**B**) Experiment performed using only **CLP** (1.0) and PMS (1.0); (**C**) Progress of reaction after 2 h (see Table 1, Entry 12); (**D**) Crude ^1^H NMR spectrum of diastereomers **DP-****4a** after 12 h (see Table 1, Entry 8).

**Figure 3 molecules-26-05921-f003:**
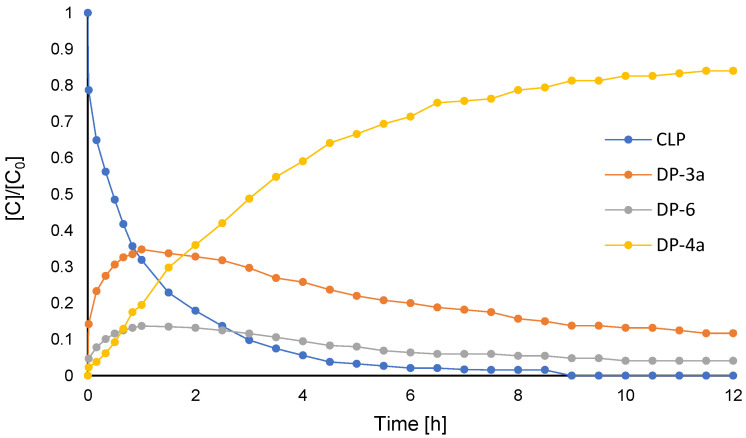
Reaction progress for oxidative chlorination reaction of **CLP** with PMS/NaCl system measured by ^1^H NMR (see Table 1, Entry 8).

**Figure 4 molecules-26-05921-f004:**
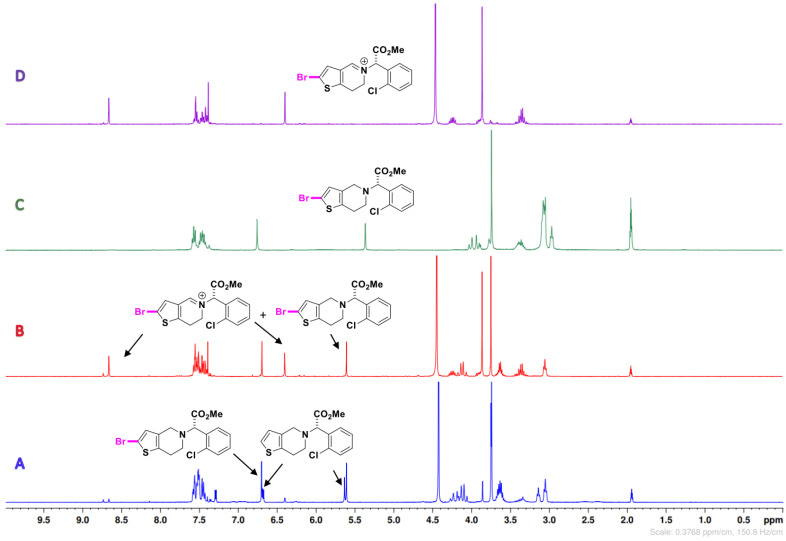
Compilation of crude ^1^H NMR spectra (400 MHz, CD_3_CN) of all compounds involved (**CLP**, **DP-3b** and **DP-5b**). Reaction conditions: (**A**) PMS (0.5), NaBr (1.0), see Table 2, Entry 6; (**B**) PMS (1.0), NaBr (1.0), see Table 2, Entry 3; (**C**) ^1^H NMR spectrum of **DP-3b** isolated; (**D**) PMS (1.5), NaBr (1.0), see Table 2, Entry 8.

**Figure 5 molecules-26-05921-f005:**
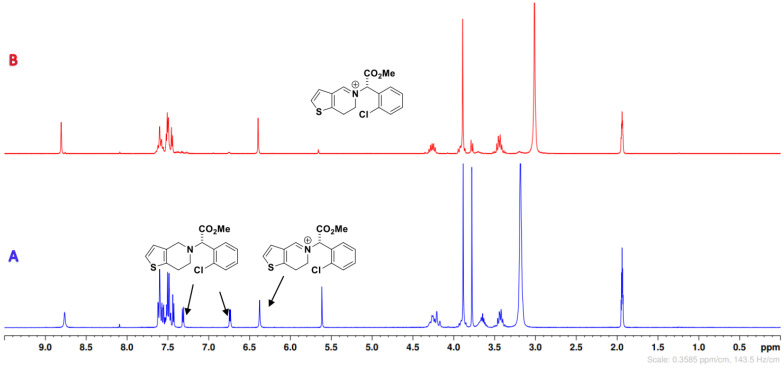
Compilation of crude ^1^H NMR spectra (400 MHz, CD_3_CN) of all compounds involved (**CLP** and **DP-2**). Reaction conditions: (**A**): PMS (0.5), NaI (1.0), see Table 3, Entry 5; (**B**) PMS (1.0), NaI (1.0), see Table 3, Entry 4.

**Table 1 molecules-26-05921-t001:**
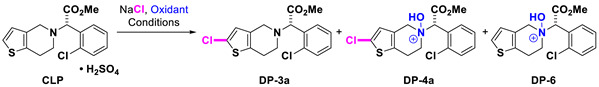
Screening of conditions for oxidative chlorination reaction ^(a)^.

Entry	NaCl(equiv)	Oxidant(equiv)	Deuterated Solvent ^(b)^	Time(h)	DP-3a(%) ^(d)^	DP-4a(%) ^(d)^	DP-6(%) ^(d)^
1	1.0	H_2_O_2_ (30%)	H_2_O_2_:CD_3_OD (2:1)	24	NR	NR	NR
2	1.0	TBHP (70%)	TBHP:CD_3_OD (2:1)	24	NR	NR	NR
3	1.0	PMS (2.0)	D_2_O ^(c)^	1 min	degrad.	degrad.	degrad.
4	1.0	PMS (2.0)	D_2_O:(CD_3_)_2_CO (2:1)	5	4	25	48
5	1.0	PMS (2.0)	D_2_O:CD_3_OD (2:1)	5	19	32	24
6	1.0	PMS (2.0)	D_2_O:C_4_D_8_O (2:1)	5	1	51	45
7	1.0	PMS (2.0)	D_2_O:CD_3_CN (2:1)	4	6	51	36
**8**	**2.0**	**PMS (2.0)**	**D_2_O:CD_3_CN (2:1)**	**4**	**4**	**85**	**10**
9	1.0	PMS (1.5)	D_2_O:CD_3_CN (2:1)	4	2	52	42
10	2.0	PMS (1.5)	D_2_O:CD_3_CN (2:1)	4	17	59	19
11	1.0	PMS (1.0)	D_2_O:CD_3_CN (2:1)	4	5	27	52
12	2.0	PMS (1.0)	D_2_O:CD_3_CN (2:1)	4	15	32	33
13	1.0	PMS (0.5)	D_2_O:CD_3_CN (2:1)	4	5	7	42

^(a)^ Reaction conditions: **CLP** (1.0 equiv), NaCl, PMS, 0.2 mL of co-solvent, 0.4 mL of D_2_O. ^(b)^ List of abbreviations: (TBHP) *tert*-butyl hydroperoxide; (CD_3_OD) methanol-d_4_; (D_2_O) water-d_2_; (CD_3_CN) acetonitrile-d_3_; ((CD_3_)_2_CO) acetone-d_6_; (C_4_D_8_O) tetrahydrofuran-d_8_; (NR) no reaction; (degrad.) degradation. ^(c)^ Only D_2_O was used (0.6 mL). ^(d)^ Conversion determined by HPLC analysis. Note: Entry 8 (in bold) refers to the best condition for the formation of **DP-4a**.

**Table 2 molecules-26-05921-t002:**
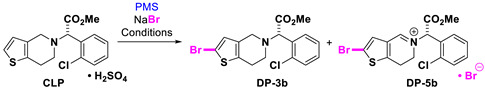
Screening of conditions for oxidative bromination reaction ^(a)^.

Entry	NaBr(equiv)	PMS (equiv)	Deut. Solvent	Time(min)	DP-3b(%) ^(b)^	DP-5b(%) ^(b)^
1	NaBr (1.0)	1.0	D_2_O:CD_3_OD (2:1)	1	62	38
2	NaBr (1.0)	1.0	D_2_O:C_4_D_8_O (2:1)	1	3	97
3	NaBr (1.0)	1.0	D_2_O:(CD_3_)_2_CO (2:1)	1	34	66
4	NaBr (1.0)	1.0	D_2_O:CD_3_CN (1:1)	1	60	39
5	NaBr (2.0)	1.0	D_2_O:CD_3_CN (1:1)	1	34	66
**6**	**NaBr (1.0)**	**0.5**	**D_2_O:CD_3_CN (1:1)**	**1**	**66**	**2**
7	NaBr (2.0)	0.5	D_2_O:CD_3_CN (1:1)	1	61	10
**8**	**NaBr (1.0)**	**1.5**	**D_2_O:CD_3_CN (1:1)**	**1**	**-**	**99**
9	NaBr (2.0)	1.5	D_2_O:CD_3_CN (1:1)	1	-	99
10	NaBr (1.0)	2.0	D_2_O:CD_3_CN (1:1)	1	-	99

^(a)^ Reaction conditions: **CLP** (1.0 equiv), NaBr, PMS, 0.2 mL of co-solvent, 0.4 mL of D_2_O. ^(b)^ Conversion determined by HPLC analysis. Note: Entries 6 and 8 (in bold) refer to the best conditions for the formation of **DP-3b** and **DP-5b**, respectively.

**Table 3 molecules-26-05921-t003:**
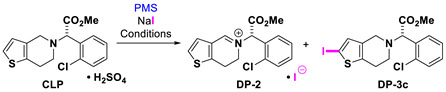
Screening of conditions for oxidative iodination reaction ^(a)^.

Entry	NaI (equiv)	PMS (equiv)	Deut. Solvent	Time	DP-2 (%) ^(b)^	DP-3c (%) ^(b)^
1	NaI (1.0)	1.0	D_2_O:CD_3_OD (2:1)	12 h	<1	-
2	NaI (1.0)	1.0	D_2_O:C_4_D_8_O (1:1)	12 h	NR	NR
3	NaI (1.0)	1.0	D_2_O:(CD_3_)_2_CO (1:1)	12 h	NR	NR
**4**	**NaI (1.0)**	**1.0**	**D_2_O:CD_3_CN (1:1)**	**10 min**	**98**	**-**
5	NaI (1.0)	0.5	D_2_O:CD_3_CN (1:1)	10 min	65	-

^(a)^ Reaction conditions: **CLP** (1.0 equiv), NaI, PMS, 0.2 mL of co-solvent, 0.4 mL of D_2_O. ^(b)^ Conversion determined by HPLC analysis. Note: Entry 4 (in bold) refers to the best condition for the formation of **DP-2**.

**Table 4 molecules-26-05921-t004:**
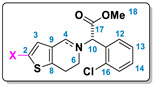
^1^H and ^13^C assignments for all clopidogrel degradation products.

Position	X = H (DP-2)		X = Cl (DP-3a)		X = Cl, N-OH (DP-4a) ^(a)^		X = Cl, N-OH (DP-4a’) ^(a)^		X = Br (DP-3b)		X = Br (DP-5a)	
	^1^H	^13^C	^1^H	^13^C	^1^H	^13^C	^1^H	^13^C	^1^H	^13^C	^1^H	^13^C
2 (CX)	-	128.15	-	133.59	-	125.72	-	125.39	-	110.38	-	128.41
3 (CH)	-	129.44	6.50 s	125.54	6.70 s	125.26	6.60 s	125.26	6.70 s	129.36	-	133.01
4 (CH/H_2_)	8.81 s	162.40	3.64 d (14.5) 3.54 d (14.5)	51.07	5.09 d (15.5) 4.82 d (15.5)	62.50	4.65 (s)	62.27	3.63 dt (14.4, 1.9) 3.53 dt (14.4, 1.9)	51.10	9.05 s	162.46
6 (CH_2_)	4.30–4.23 m 3.95–3.86 m	49.94	2.98–2.84 m	49.21	4.22 dd (12.1, 5.3) 4.03 m	60.68	4.40 m 4.29 dd (9.2, 5.1)	61.99	2.95–2.80 m	49.21	4.44–4.37 m 4.00–3.91 m	49.64
7 (CH_2_)	3.52–3.36 m	24.04	2.76–2.70 m	25.78	3.04 m	21.54	3.24 m	22.19	2.77–2.74 m	26.02	3.59–3.40 m	24.38
8 (C)	-	156.50	-	128.22	-	130.56	-	130.48	-	136.39	-	158.09
9 (C)	-	129.02	-	133.28	-	130.39	-	130.18	-	135.32	-	130.19
10 (CH)	6.40 s	72.08	4.97 s	68.55	6.19 (s)	76.95	6.15 (s)	74.93	4.93 s	68.68	6.62 s	72.72
11 (C_(ipso)_)	-	129.35	-	134.13	-	134.58	-	134.55	-	134.76	-	115.56
12 (C_Ar_H)	7.62–7.49 m	133.81	7.44–7.41 m	130.99	7.65–7.53 m	129.20	7.65–7.53 m	129.09	7.48–7.43 m	131.02	7.68–7.56 m	131.29
13 (C_Ar_H)	-	129.16	7.33–7.29 m	128.48	7.65–7.53 m	134.58	7.65–7.53 m	134.54	7.37–7.31 m	128.50		129.61
14 (C_Ar_H)	-	132.86		131.11	7.50–7.43 m	131.91	7.50–7.43 m	131.83		131.05		133.01
15 (C_Ar_H)	-	131.83	7.63–7.60 m	130.99	7.73 dd (7.9, 1.6)	133.93	7.77 dd (7.9, 1.6)	133.93	7.66–7.61 m	131.02		132.08
16 (CCl)	-	136.18	-	135.80	-	137.59	-	137.17	-	135.91	-	136.54
17 (C=O)	-	167.81	-	172.10	-	166.27	-	166.17	-	172.76	-	168.20
18 (OCH_3_)	3.89 s	55.02	3.68 s	52.84	3.75 (s)	55.06	3.74 (s)	55.06	3.70 s	52.71	3.97 s	54.85

Note: Positions show the number on the chemical structure above. Chemical shifts (δ) are reported in ppm relative to TMS (see Section 3.2). *J*- values are shown in Hz in parentheses. Abbreviations: d, doublet; dd, doublet doublet; dt, doublet triplet; m, multiplet; s, singlet; t, triplet. ^(a) 1^H NMR (400 MHz, CD_3_CN) signals correspond to 56:44 mixture of diastereomers **DP-4a** and **DP-4a’** (cf. ref. [37]).

## Data Availability

The data presented in this study are available in Appendix A.

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
