# Peer review of "Selective Oxidation of Clopidogrel by Peroxymonosulfate (PMS) and Sodium Halide (NaX) System: An NMR Study"

_molecules, 2021, doi:10.3390/molecules26195921_

Round 1

Reviewer 1 Report

This is an excellent investigation tha cam shed light on how Clopidrogel degrades.

Some of the improvements that can be made to the document are:
1. In Figure 1D, the authors must say what are the counterions of compounds DP-2, DP-4a and DP-5b, respectively.
2. In line 83, what the authors said does not correspond to what is presented in Table 1 (entries 10-12).
3.
In the description of the experimental part, the authors must specify the different amounts of solvent mixtures (D2O / CD3CN) used to carry out the reactions.

Author Response

Reviewer #1: This is an excellent investigation that can shed light on how Clopidogrel degrades. Some of the improvements that can be made to the document are:

  1. In Figure 1D, the authors must say what are the counterions of compounds DP-2, DP-4a and DP-5b, respectively. Done, an explaining comment has been added.
  2. In line 83, what the authors said does not correspond to what is presented in Table 1 (entries 10-12). Done. The clarifying corrections are highlighted in yellow (lines 83, 85 and 88).

3.In the description of the experimental part, the authors must specify the different amounts of solvent mixtures (D2O / CD3CN) used to carry out the reactions. Done, an explanatory statement has been added, line 294. The corrections are highlighted in yellow on Supporting Information (SI, pages S16 ‒ S20).

Reviewer 2 Report

REVIEW MOLECULES 1368882

Selective Oxidation of Clopidogrel by Peroxymonosulfate 2 (PMS) and Sodium Halide (NaX) System: an NMR Study

Everaldo F. Krake and Wolfgang Baumann *

The current manuscript is a well structured, well written and very interesting work that focuses on a very relevant issue such as drug reactivity and the generation of impurities that could have some biological implication. I recommend accepting the article after making some minor corrections.

Minor corrections

Page 1, line 30-32. Reference is needed.

Page 1, line 43. Please change specie by species

Page 2, line 72.  Please define TBHP in the text.

Along the text. Please revise structures with N-oxide function. Traditionally, a N-oxide is represented as:  R3N(+)O(-) and not as: R3N(+)OH                  

Author Response

Reviewer #2: The current manuscript is a well-structured, well written and very interesting work that focuses on a very relevant issue such as drug reactivity and the generation of impurities that could have some biological implication. I recommend accepting the article after making some minor corrections.

Minor corrections

Page 1, line 30-32. Reference is needed. Done. This topic is also addressed in lines 39/40, references 8 and 9.

Page 1, line 43. Please change specie by species. Done

Page 2, line 72. Please define TBHP in the text. Done. The definition about TBHP was also mentioned before on the Table 1 (see footnote (b): list of abbreviations).

Along the text. Please revise structures with N-oxide function. Traditionally, a N-oxide is represented as: R3N(+)O(-) and not as: R3N(+)OH. We are aware that a N-oxide is represented as R3N(+)O(-), but in protic media the representation R3N(+)OH is also acceptable for this species.

Others groups also used this representation for N-oxide as R3N(+)OH, see: (a) Baertschi et al, J. Pharm. Sci. 2019, 108, 2842−2857; (b) O’Shea et al, J. Hazard. Mater. 2020, 398, 123219 and us (c) Krake et al, J. Mol. Struct. 2022, 1247, 131309 (DOI: https://doi.org/10.1016/j.molstruc.2021.131309).

Reviewer 3 Report

The selective introduction of halogen atoms into heteroaromatic rings including heterocycle-containing drugs and compounds with biological activity is an important task for organic chemists. The presented paper describes a selective transformation of clopidogrel hydrogen sulfate by reactive halogen species generated from peroxymonosulfate  and sodium halide. Other sustainable oxidants as well as different solvents have also been investigated.  As result of this research, the reaction conditions were optimized for sodium chloride, bromide and iodide, and a number of degradation products and halogen-containing derivative of clopidogrel were studied.

In my opinion, the results obtained are important not only as studies of the transformations of the currently used drug, but also as the development of sustainable methods for selective introduction of halogen atoms into heteroaromatic rings as well as the synthesis of new derivatives with promising biological activity.

The article is written in good scientific language, well-framed and demonstrates a varied spectral basis. I did not find any noticeable shortcomings and suggest that the article can be accepted in the present form.

Author Response

No changes were made here.

We thank the reviewer for the analysis and comments on this work.